# LEARNING GENERAL AND REUSABLE FEATURES VIA RACECAR-TRAINING

## ABSTRACT

We propose a novel training approach for improving the learning of generalizing features in neural networks. We augment the network with a reverse pass which aims for reconstructing the full sequence of internal states of the network. Despite being a surprisingly simple change, we demonstrate that this forward-backward training approach, i.e. *racecar* training, leads to significantly more general features to be extracted from a given data set. We demonstrate in our paper that a network obtained in this way is continually trained for the original task, it outperforms baseline models trained in a regular fashion. This improved performance is visible for a wide range of learning tasks from classification, to regression and stylization. In addition, networks trained with our approach exhibit improved performance for task transfers. We additionally analyze the mutual information of our networks to explain the improved generalizing capabilities.

## 1 INTRODUCTION

Humans spend surprising amounts of time assembling and disassembling objects. These deconstruction tasks serve a wide range of purposes from a hobby for motorists, to an important source of learning and exploration for children (Gopnik et al., 1999). Motivated by this behavioral trait of humans, we propose a surprisingly simple, yet powerful modification of neural network training: in addition to a regular forward pass, we add a reverse pass that is constrained to reconstruct all in-between results of the forward pass as well as the input. As we will demonstrate below, this palindromic structure yields substantial improvements for generalization of the learned features in a wide range of architectures, and in many cases even improves the baseline achieved with regular training. Our results indicate that the reversible nature of the proposed training setup, which we will subsequently refer to via the palindrome *"racecar"*, encourages the formation of general and reusable features that benefit a wide range of learning tasks.

With our approach we specifically target transfer learning applications. For a regular, i.e., a non-transfer task, the goal usually is to train a network that gives the optimal performance for one specific goal, as has been demonstrated in many success stories over the years (LeCun et al., 1998; Krizhevsky et al., 2012; Goodfellow et al., 2014; He et al., 2016). In such a case, the network naturally exploits any observed correlations between input and output distribution. E.g., if the color of an object in any way correlates with its type, the training of a classifier should find and use this information. In recent years, even networks trained only for a very specific task were shown to be powerful starting points for training models with different tasks (Zamir et al., 2018; Gopalakrishnan et al., 2017; Ding et al., 2017). In many cases, the original network contained features that were applicable to different data domains and beneficial for new inference tasks. An inherent difficulty in this setting is that typically no knowledge about the specifics of the new data and task domains is available at training time of the source model. While it is common practice to target broad and difficult tasks with the hope that this will yield learned features that are applicable in new domains, we instead specifically target improving the generalizing capabilities of learned features while training the source model.

The core idea of our approach is to add a reverse path during training that is constrained to be as reversible as possible every step of the way. This makes the training task more difficult at first, but at the same time encourages the network to learn reversible and general features. Motivated by the disassembly and assembly processes of humans, and in contrast to previous work on invertible

networks (Gomez et al., 2017; Jacobsen et al., 2018; Zhang et al., 2018), we constrain the network to represent a as-reversible-as-possible process for all intermediate layer activations, instead of only perfectly reproducing the input. Thus, even for cases where a classifier can, e.g., rely on color for inference of an object type, the model is encouraged to learn a representation that can recover the input in order to not only reconstruct the color of an object but also its shape. Hence the internal representation the network builds naturally has to encode as many aspects of the input data distribution as its representational capabilities permit, in order to recover the input. We demonstrate the benefits of our approach for a variety of architectures, from pure convolutional neural networks (CNNs) with and without batch normalization, to networks that include fully connected layers, as well as GAN architectures.

## 2 RELATED WORK

Transfer learning with deep neural networks has been very successful for a variety of tasks, such as image classification (Duan et al., 2012; Kulis et al., 2011; Zhu et al., 2011), multi-language text classification (Zhou et al., 2014b; Prettenhofer & Stein, 2010; Zhou et al., 2014a), and medical imaging problems (Ravishankar et al., 2016). A central question in transfer learning is whether base and target tasks are related or not. Zamir et al. (2018) proposed an approach to obtain task relationship graphs for different tasks. Another critical point is that we need the neural networks to learn general features from the data set, which are useful for both base and related tasks, rather than some specific features. But how to improve generalization of trained neural networks and reuse them for different related tasks is still a challenge.

In this paper we propose a modified training approach for improving generalization via explicitly building a reverse pass network in addition to a regular forward pass. While Zhang et al. (2018) proposed reverse connected modules, they are primarily used to transfer information from deep layers to shallow layers for a given task. Recovering all input information from hidden representations of a network is generally very difficult (Dinh et al., 2016; Mahendran & Vedaldi, 2016), due to the loss of information over the course of the layer transformations. In this context, Tishby & Zaslavsky (2015) proposed the information bottleneck principle, which states that for an optimal representation, information unrelated to the current task is reduced. This highlights the common specialization of regular training approaches. Ardizzone et al. (2018), Jacobsen et al. (2018) and Gomez et al. (2017) also build reversed networks, but mainly focus on how to make a network fully invertible via introducing special structures. As a consequence, the path from input to output is actually different from the reverse path that translates output to input. In addition, the proposed structures of previous work can not be easily attached to other network architectures. In contrast, we show that it is not necessary to strive for perfect reversibility to obtain an improved performance. Our racecar training also fully preserves an architecture for the backward path, and does not require operations that are not part of the source network. As such it can easily be applied in new settings, such as for adversarial training (Goodfellow et al., 2014).

Aiming for related goals, Bansal et al. (2018) introduced orthogonality regularizations to the loss function. However, the proposed constraints are relatively weak, and make it difficult to arrive at invertible networks. Besides, the orthogonality regularization still focuses on improving performance of a known, given task. This means the training process only extracts features which the network considers useful for improving the performance of the current task. Hence, unlike our method, orthogonality does not necessarily improve generalization or improved transfer performance (Torrey & Shavlik, 2010).

## 3 METHOD

The goal of transfer learning is to reuse a basic model trained for task A for a related new task B. The performance for B naturally depends crucially on whether the content of the basic model yields benefits for the new task. In line with previous work (Yosinski et al., 2014), we consider the generality of features learned on task A as the extent to which the features can be used for task B. This can be measured in terms of the difference between the transfer performance $p_{AB}$ and training a model for task B from scratch yielding performance $p_B$. Our central goal is improving the

*transferability* of the basic model w.r.t. the performance metric $p_{AB} - p_B$, which at the same time can be seen as a measure of the generalizing capabilities of the basic model.

Regular training approaches typically construct a model with a certain network structure, and train the model weights for a given task via a suitable loss function. Our approach does not modify this initial structure, but adds a second pass that reverses the initial structure while reusing all weights and biases. E.g., for a typical fully connected layer, the forward pass, the operation $L_2 = M \times L_1 + b$ is changed to $L_1' = M^T \times (L_2 - b)$ for the reverse pass, where $L_1$ and $L_2$ denote input and output, respectively. Here, $M$ and $b$ denote weight matrix and bias, while $L_1'$ denotes the input regenerated via the reverse pass. We will show and discuss $L'$ for several examples below.

Our goal with the reverse pass is to invert all operations of the forward pass to obtain identical intermediate activations between the layers with matching dimensionality. We can then constrain the intermediate results of each layer of the forward pass to match the results of the backward pass. Due to the symmetric structure of the two passes, we can use a simple $\mathcal{L}^2$ difference to drive the network towards aligning the results:

$$\mathcal{L}_{\text{racecar}} = \sum_{m=1}^{n} \lambda_m \left\| L_m - L_m' \right\|_2 \tag{1}$$

Here $L_m$ denotes the input of layer $m$ in the forward pass and $L_m'$ the output of layer $m$ for the reverse pass. $\lambda_m$ denotes a scaling factor for the loss of layer $m$, which, however, is typically constant in our tests across all layers. An illustration of this process for a CNN structure is shown in Fig. 1. While the construction of the reverse is straight-forward for all standard operations, i.e., fully connected layers, convolutions, pooling etc., batch normalization (BN) and activation function require slight adjustments to map $L_m$ and $L_m'$ to the same range of values such that they can be compared in the loss. Hence, we use the BN parameters and the activation function of layer $m - 1$ from the forward pass for layer $m$ in the reverse pass, Note that with our notation, $L_1$ and $L_1'$ refer to the input $I$, and the regenerated input $I'$, respectively. In the following, we will refer to networks trained with the added reverse structure and the loss terms of equation 1 as *racecar training*.

The constraints of equation 1 intentionally only minimize differences in an averaged manner with an $\mathcal{L}^2$ norm, as we don't strive for a perfectly bijective mapping between input and output domains. Rather, our goal with the racecar training is to encourage the network to extract features that preserve as much information from the input data set as possible with the given representative capabilities of the chosen architecture. Hence, while a regular forward training allows a model to specialize its extracted features to maximize performance for a given task, our approach encourages the network to consider the full input data distribution, such that it ideally can be recovered from the latent-space representation of all layers.

To differentiate variants, we will use the following naming scheme: $\text{Std}_{A/B}$, $\text{RR}_{A/B}^s$, $\text{Ort}_{A/B}$ for trained base models. Here, $Std$ denotes a regular training run (always shown in yellow color in graphs below), while $RR^s$ denotes models trained with our racecar training (in green below). Here, the $s$ superscript denotes how many intermediate results of layers are used, e.g., $RR^3$ means $n = 3$ in equation 1, which constrains the input data as well as the next two layers of the original structure.

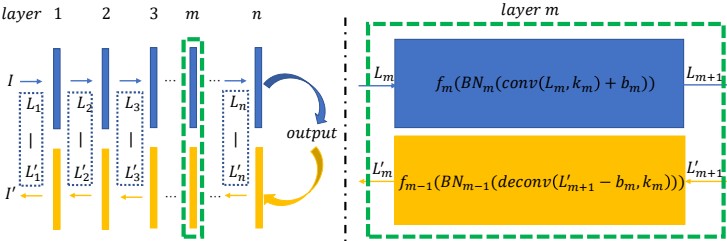

Figure 1: Left: An overview of the regular forward pass (blue) and corresponding reverse pass (yellow). The right side illustrates how parameters are reused for a convolutional layer. *conv* and *deconv* denote convolution and deconvolutional operations, and $f_m$ and $BN_m$ denote activation function and batch normalization of layer $m$, respectively. Shared kernel and bias are represented by $k_m$ and $b_m$, and we assume that the deconvolution internally transposes the kernel tensor.

Correspondingly, a network trained with $RR^1$ means the network is trained to regenerate the input without any constraints for the activations inside of the network. This represents a special, and somewhat sub-optimal case of our method, as we will demonstrate below. $Ort$ additionally denotes models trained with orthogonal constraints (Bansal et al., 2018) (in blue). The subscripts $A/B$ denote the task $A/B$ the model was trained for. We will call direct training for either task phase I in the following.

In phase II, we reuse a model trained in phase I for new tasks (typically B), with a regular training approach. I.e., in phase II, all models are only trained with the forward structure and original loss, and do not use the racecar loss or other modifications. We will use the naming scheme $\text{Std}_{\text{AA/AB}}$, $\text{RR}^s_{\text{AA/AB}}$, $\text{Ort}_{\text{AA/AB}}$ for trained models in phase II. Here $AA/AB$ mean the model was trained for task $A$ during phase I, and is then trained for task $A/B$ in phase II.

It is worth pointing out that the additional constraints of our racecar training lead to increased requirements for memory and additional computations during phase I, e.g., it is 61.13% slower per epoch for the MNIST tests. And while the constraints can also lead to a slight deterioration of performance, e.g., a 0.17% lower accuracy for MNIST tests during phase I, we will demonstrate that our models outperform baselines during phase II even for the original task, and thus in practice justify introducing the racecar training.

# 4 Evaluation in Terms of Mutual Information

We now evaluate our approach in terms of the mutual information (MI) between input and output data distributions. As our approach hinges on the introduction of the reverse pass, we will show that our approach succeeds in terms of establishing mutual information between the input and the constrained intermediates inside a network. More formally, the mutual information $I(X;Y)$ of random variables $X$ and $Y$ measures how different the joint distribution of $X$ and $Y$ is w.r.t. the product of their marginal distributions, i.e., the Kullback-Leibler divergence: $I(X;Y) = D_{KL}[P_{(X,Y)}||P_X P_Y]$.

Tishby & Zaslavsky (2015) proposed a *mutual information plane* to analyze the training process, which shows $I(X;L)$ and $I(L;Y)$ for each layer $L$ over the course of the training epochs. Points in the resulting graphs are colored w.r.t. the epoch, i.e., initially black, and yellow once the training is finished. Each graph shows the 5 middle and output layers. The MI planes visualize how much information about input and output distribution is retained at each layer, and how these relationships change within the network. For regular training, the information bottleneck principle (Tishby & Zaslavsky, 2015) states that early layers contain more information about the input (high $I(X;L)$ and $I(Y;L)$). Hence they are often visible at the top-right. Later layers have no relationship with the output $I(Y;L)$ initially, which increases over the course of a successful training. Thus, they typically move

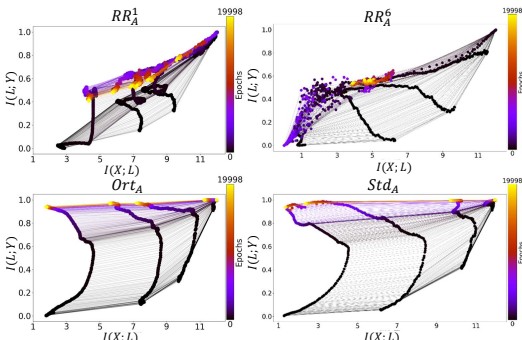

Figure 2: MI planes for the initial models $\text{RR}^1_A$, $\text{RR}^6_A$, $\text{Std}_A$ and $\text{Ort}_A$. End points of $\text{RR}^6_A$ are located in the middle of the graphs. But for $\text{Std}_A$ and $\text{Ort}_A$, $I(L;Y)$ values of every layer are very high, which means more specific features about the given output data set are learned by them.

to the top-left. The intuition behind MI planes is explained in more detail in Appendix A.1. We use the same numerical studies as in Shwartz-Ziv & Tishby (2017) as task $A$, i.e. a regular feed-forward neural network with 6 fully-connected layers. The inputs consist of 12 binary digits, and outputs are 2 binary digits. Models are trained using cross entropy as base loss function. Details of this architecture (and following ones) are given in the Sec. A.1. At first, we train a base model $\text{RR}^6_A$ with a full racecar loss. For comparison, we also show the MI planes for $\text{Std}_A$ (a regularly trained model), and $\text{Ort}_A$ (trained with orthogonal constraints (Bansal et al., 2018)). We additionally include a version $\text{RR}^1_A$, i.e. trained with only one racecar loss term $\lambda_1 |L_1 - L'_1|_2$), which means that only the input is constrained to be recovered. Thus, $\text{RR}^1_A$ represents a sim-

plified version of our approach which receives no constraints that the intermediate results of forward and backward pass should match. The resulting information planes are shown in Fig. 2. We can see that the end points of $RR_A^6$, i.e., the yellow clusters of points for the final state of the model, are located in the middle part of the graph. This means that all layers successfully encode information about the inputs as well as the outputs. In contrast, for $Std_A$ and $Ort_A$, $I(X;L)$ values of later layers are very low (points near the top left), while $I(L;Y)$ is high throughout. This indicates that the outputs were successfully encoded, and that increasing amounts of information about the inputs are discarded by $Std_A$ and $Ort_A$. Hence, both models are highly specialized for the given task. Comparing $RR_A^1$ with $RR_A^6$, the yellow end points of the latter are located in a centralized region, which indicates that every layer contains similar information about $X$ and $Y$. It also indicates that the path from input to output is similar to the path from output to input. The end points of $RR_A^1$, on the other hand, are located in a scattered region. I.e., this network has different amounts of mutual information across its layers, and potentially a very different path in each direction. $RR_A^1$ is only constrained to be able to regenerate its input, while the full racecar loss for $RR_A^6$ ensures that the network learns features which are beneficial for both directions. This test highlights the importance of the constraints throughout the depth of a network in our racecar loss.

As the discussion above focused on phase I, we now turn to analyzing phase II, i.e., follow-up training runs with models trained during phase I. Based on $RR_A^1$, $RR_A^6$, $Ort_A$ and $Std_A$, we continue training the models only for the original task, i.e., $RR_{AA}^1$, $RR_{AA}^6$, $Ort_{AA}$ and $Std_{AA}$. The resulting MI planes are shown in Fig. 3. While all models now focus on the output (yellow points at the top, maximizing $I(L;Y)$), there are differences in the distributions of the yellow points along the

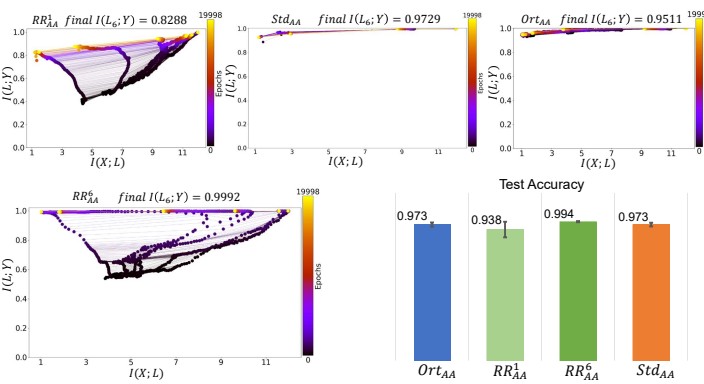

Figure 3: MI plane and accuracy comparisons for continued training ($RR_{AA}^1$, $RR_{AA}^6$, $Std_{AA}$ and $Ort_{AA}$). The $RR_{AA}^6$ model slightly outperforms the other three models, indicating the positive effect of the full racecar training.

x axis, i.e., how much MI with the input is retained. We can see that for model $RR_{AA}^6$, the final $I(X;L_6)$ value at the end of the training is higher than for $Std_{AA}$, $Ort_{AA}$ and $RR_{AA}^1$ (final $I(L_6;Y)$ values are shown above each graph). While the final accuracy is high throughout due to the relatively simple setup, $RR_{AA}^6$ slightly outperforms the other variants for the original task. We will more clearly demonstrate this positive trait of our racecar training with more complex tasks below.

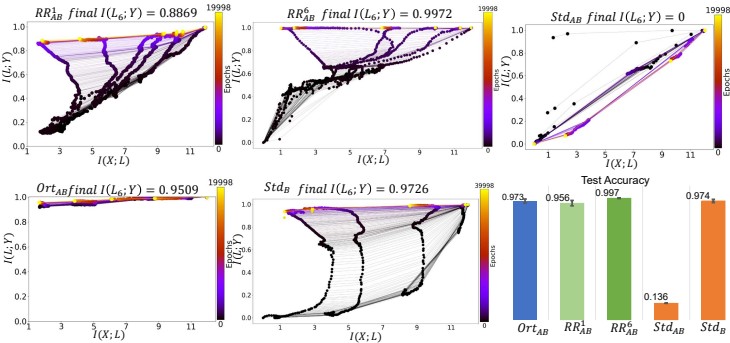

Figure 4: MI plane and accuracy comparisons for a task transfer for models $RR_{AB}^1$, $RR_{AB}^6$, $Ort_{AB}$ and $Std_{AB}$. $RR_{AB}^6$ successfully reuses features from task $A$ for task B, and outperforms $Std_B$. The regular model $Std_{AB}$ yields a very low performance.

We now analyze a new task, to check whether the model learned specific or general features, and reverse output labels as transfer learning task $B$. E.g., if the output is [0,1] in the original data set, we invert it to [1,0]. Based on $RR_A^1$, $RR_A^6$, $Ort_A$ and $Std_A$, we train $RR_{AB}^1$, $RR_{AB}^6$, $Ort_{AB}$ and $Std_{AB}$ for the modified data set. For comparison, we also train a model trained $Std_B$ from scratch. MI planes and accuracy comparisons are shown in Fig. 4. While the range in performance is relatively small across most models, $Std_{AB}$ stands out with a very low accuracy. This model from a regular training run has large difficulties to adapt to the new task. Model $Ort_{AB}$ also performs worse than $Std_B$. $RR_{AB}^6$ shows the best performance in this setting, confirming the previous MI plane visualizations, and demonstrating that our loss formulation helped to learn more general features from the input data, improving the performance for related tasks such as the inverted outputs.

## 5 EXPERIMENTAL RESULTS

We now turn to more complex network structures (CNNs, GANs, Auto-Encoders), with different data sets (MNIST, Cifar, smoke, ImageNet) and different tasks (such as classification and synthesis) to show that models trained with our approach succeed in learning very general features that transfer to new tasks.

### 5.1 DIGIT CLASSIFICATION

The MNIST data set is a commonly used data set for hand written digit classification (LeCun et al., 1998). At first, we train three models for regular MNIST data set classification as task $A$, one with racecar loss $RR_A^3$, a regular model $Std_A$, and one with orthogonal $Ort_A$ constraints (Bansal et al., 2018), as the latter also aims for improving CNN performance. Usually, convolutional layers take up most of the model parameters, so we correspondingly compute the racecar loss for the convolutional layers, omitting the fully connected layers that would be required for class label inference. To

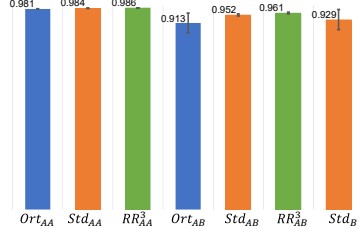

Figure 5: Comparisons between regenerated inputs. F.l.t.r.: $Ort_A$, $Std_A$, $RR_A^3$, and the reference. Only $RR_A^3$ recovers most of the input information successfully.

highlight the properties of our algorithm, we show comparisons between $I$ and the regenerated $I'$ in Fig. 5. We can see that for racecar training, most of the features from the input are recovered. Trying to invert the network in the same way for a regular training run or training with orthogonal constraints largely fails, as the extracted features are extracted according to the digit classification and discard information unrelated to this task.

Based on $Ort_A$, $Std_A$, $RR_A^3$, we continue training to obtain models $Ort_{AA}$, $Std_{AA}$, and $RR_{AA}^3$. Results are shown in Fig. 6. We repeated these runs 5 times, in order to draw reliable conclusions. We can see that $RR_{AA}^3$ outperforms $Ort_{AA}$ and $Std_{AA}$ at the end of training, which indicates that racecar training yields general features that can also improve performance for original task. Council (2000) illustrated that balanced learning of both general and specific features is more effective for human learning, and our results here are consistent with this intuition.

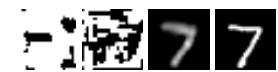

Figure 6: Accuracy comparisons for base task $A$ and transfer learning task $B$. $RR_{AA}^3$ and $RR_{AB}^3$ achieve the best performance for the base task and transfer learning task, respectively.

As general features are more robust than specific features (Novak et al., 2018), we investigate a perturbed data set for the transfer task B. We apply $Ort_A$, $Std_A$ and $RR_A^3$ to n-MNIST, a data set for classification with motion blur (Basu et al., 2017). Performance results are likewise given in Fig. 6. Based on the same CNN structure and parameters, $RR_A^3$ achieves the best performance. This indicates that $RR_A^3$ learned more general features via racecar training than $Ort_A$ and $Std_A$.

### 5.2 NATURAL IMAGE CLASSIFICATION

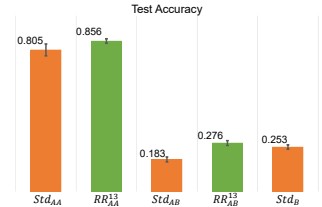

Natural images arise in many important application scenarios. Hence, we evaluate our approach with the Cifar data set (Krizhevsky et al., 2009). At first, we train two models as Cifar-10 data set classification task $A$, $RR_A^{13}$ and $Std_A$. We again continue training to obtain $RR_{AA}^{13}$ and $Std_{AA}$, results for which are shown in Fig. 7. The racecar training also improves performance for this natural image classification task. For transfer learning task $B$, we reuse $RR_A^{13}$ and $Std_A$ for Cifar-100 data set classification ($RR_{AB}^{13}$ and $Std_{AB}$). Results are also shown in Fig. 7. Training with the same CNN structure and parameters, $Std_{AB}$ has difficulties adjusting to the new task, while our model from the initial racecar training slightly outperforms the model trained for scratch for B.

Figure 7: Accuracy comparisons of task $A$ and task $B$. $RR_{AA}^{13}$ and $RR_{AB}^{13}$ got bestperformance for tast $A$ and $B$.

### 5.3 GENERATIVE ADVERSARIAL MODELS

In the previous sections, we employed commonly used data sets and transfers to related tasks. Next, we test our approach for a transfer situation with a more challenging transfer from a synthetic single channel data set of synthetic smoke simulations, to RBG videos of real smoke clouds. We use GAN and auto-encoder structures for the following models.

At first, we use a GAN structure (one generator and one discriminator network) for super resolution of the simulation data as task $A$. $Std_A$ is trained via regular training, and $RR_A^6$ uses racecar training for the generator network of the GAN. Regenerated low resolution results and high resolution outputs comparisons are shown in Fig. 8. Both $RR_A^6$ and $Std_A$ can up scale low resolution data 4 times larger to high resolution data very well. But $RR_A^6$ can also recover low resolution versions from high resolution data. $Std_A$ fails in regenerating the low resolution images. This shows that $RR_A^6$ not only focused on

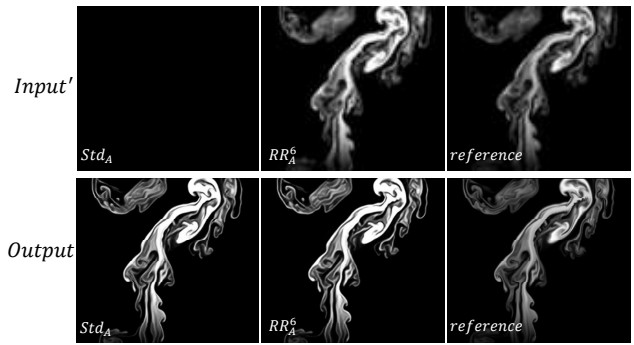

Figure 8: Regenerated low resolution results and high resolution outputs comparison between $Std_A$, $RR_A^6$ and reference. Only $RR_A^6$ successfully recovers the input.

generating good features for the super-resolution task, but also preserves information about the input.

We reuse the generator models $RR_A^6$ and $Std_A$ to train two sets of auto-encoder networks: once for the synthetic smoke data as transfer task $B_1$, and as a second task $B_2$ for RBG videos of real-world smoke clouds (example frames are given in the appendix). This yields models $RR_{AB_{1,2}}^6$ and $Std_{AB_{1,2}}$, respectively. The resulting accuracies, summarized in Fig. 9, show that racecar training performs best for both auto-encoding tasks. This is especially encouraging for task $B_2$, as it represents a transfer from fully synthetic training to real-world images.

Synthetic data and real-world images have significant differences in terms of their features. In the worst case, most of the features learned by the regular learning process $Std_A$ can not be transferred to the new task. Besides, an inappropriate starting point of the transfer learning task can guide the whole training process towards a wrong direction. Instead, a model with more general features such as $RR_A^6$ can more easily be reused in new tasks, and can provide a better starting point for learning. This indicates the potential of racecar training to obtain generalizable features from synthetic data sets that can be used for tasks working with real-world data.

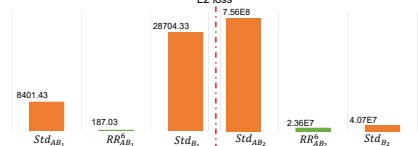

Figure 9: Accuracy comparisons of transfer learning tasks. $RR_{AA}^6$ and $RR_{AB}^6$ got best performance for task $B_1$ and $B_2$.

## 5.4 VGG19 STYLIZATION

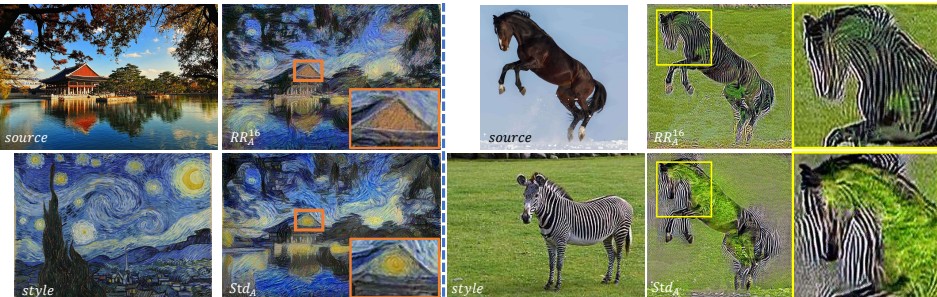

Figure 10: Left: stylization test from a natural image to the starry night painting style. Right: stylization test from horse to zebra.

In order to provide a qualitative evaluation in a complex, visual scenario, we turn to the popular task of image stylization. We use $VGG19$ networks (Simonyan & Zisserman, 2014) in the following, and we train two $VGG19$ networks $RR_A^{16}$ and $Std_A$ with ImageNet data set (Deng et al., 2009). Based on these, we train $RR_{AA}^{16}$ and $Std_{AA}$, top5 accuracy of which are $77.59\%$ and $75.28\%$, respectively. Consistent with previous test, $RR_{AA}^{16}$ outperforms the $Std_{AA}$ baseline even in this complex case for a net-

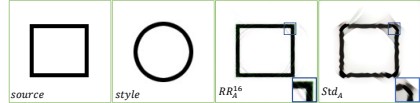

Figure 11: First primary stylization test, $RR_A^{16}$ did not change specific features in the results.

work with ca. 143 million weights. Gatys et al. (2016) achieve excellent stylization results, but the results strongly depend on the pre-trained VGG network, i.e., it stylizes and changes structures that are represented by the features in the VGG network. Hence, we employ stylization to visualize which features, general or specialized, the two VGG versions focus on.

Gatys et al. (2016) generate stylized results via optimizing two losses, $L_{style}$ and $L_{content}$, which measure style difference and content difference, respectively (details are given in the appendix. To compare the feature extracting capabilities between the $RR_A^{16}$ and $Std_A$ runs, we only optimize $L_{style}$. As a first test, we use two simple shapes as source and style images, as shown in Fig. 11. They have the same background and object color. The only difference is the large-scale shape of the object. Thus, as we aim for applying an style that is identical to the source, we can test whether the features can cleanly separate and preserve the large scale shape features from the ones for the localized style. Comparing the results of $RR_A^{16}$ and $Std_A$, the output of $RR_A^{16}$ is almost identical to the input, while stylization with $Std_A$ changes the shape of the object introducing undesirable streaks around the outline. This result indicates that $RR_A^{16}$ is able to cleanly encode the shape of the object and preserve it during the stylization.

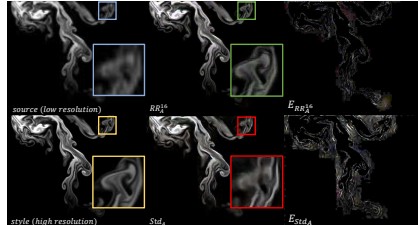

Figure 12: Stylization comparison from low resolution to high resolution. $RR_A^{16}$'s results are closer to the high resolution.

We show more stylization tests in Fig. 10, which optimize both $L_{content}$ and $L_{style}$. For the left example of Fig. 10 with a Van Gogh style image, both models can generate good stylized results. However, from the results obtained with $Std_A$ (orange square area), we can see a star within the building, which indicates that this model mixed style and context information in its features. Across our tests we found that models like this one, trained for more specific features, more often performed simple patch-based matches rather than generating meaningful transfers of styles. However, if the model focuses on more general and abstract features, it can perform a better semantic matching and generate improved stylization results. For the right part of Fig. 10, we show a test transferring a horse image to a zebra style. $RR_A^{16}$ focuses more on general features, such as *grass*, *zebra*, and their positional relationships. As a result the model successfully distinguishes foreground from background, and changes the style for both foreground and background objects with only smaller

mistakes. $Std_A$'s performance is significantly worse, leading to large grass-like areas within the body of the horse.

We confirm the capabilities of the VGG model from race-car training with a low- to high-resolution transfer where the goal is purely to add detail. I.e., we can compute in place errors w.r.t. high-resolution reference. The result is shown in Fig. 12. The VGG model obtained with a regular training run yields significantly higher errors, as visible on the right side of the figure.

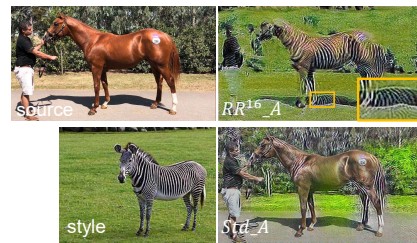

Figure 13: $RR_A^{16}$ generates a zebra pattern in the horse's shadow, but yields generally improved results compared to the regular transfer in $Std_A$.

$RR_A^{16}$ can extract more general information from the input, which is helpful with generating semantically meaningful results. However, we found that this property can also confuse the model in some specific situations. For instance, as shown in Fig. 13, at first, $RR_A^{16}$ can distinguish foreground and background and generate better stylization results than $Std_A$. However, $RR_A^{16}$ also stylizes the shadow of the horse. This also indicates that $RR_A^{16}$ focuses more on general features. As a consequence, all horse shape objects are extracted and then stylized with zebra textures, such as the shadow in Fig. 13. However, we generally think that this can provide potentially helpful information in other situations, e.g., for classification tasks. Overall, the quality of the racecar transfer also still surpasses the regular version.

## 6    CONCLUSION

We have proposed a novel training approach for improving neural network generalization by adding a constrained reverse pass. We have shown for a range of tasks that this yields networks with more general features that more easily transfer to new tasks, and even outperform baselines for the original task. Our training approach is very general, and imposes no requirements regarding network structure or training method. As future work, we believe it will be very interesting to evaluate our approach for other architectures from the vast zoo of existing work, e.g., we are particularly interested in analyzing recurrent structures such as LSTMs and GRUs (Hochreiter & Schmidhuber, 1997; Cho et al., 2014).

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

## A APPENDIX

Here we will present more details, such as data set information, network structures and training parameters, for all of our mentioned tests above: Mutual Information Sec. A.1, MNIST classification Sec. A.2, Cifar Sec. A.3, smoke Sec. A.4 and VGG Sec. A.5. We will use $C(k, l, s)$, $D(k, l, s)$ to represent convolutional and deconvolutional operations, respectively, and fully connected layers are noted with $F(l)$, where $k, l, s$ denote kernel size, output channels and stride size, respectively. Bias of CNN layer is denoted with $b$. $I/O(z)$ denote $input/output$ and their dimensionality is given by $z$. $I_r$ denotes the input of reverse pass network. $tanh$, $relu$, $lrelu$ denote corresponding activation functions, where we typically use a leaky tangent of 0.2 for the negative half space. $UP$, $MP$ and $BN$ denote $2\times$ nearest-neighbor up sampling, max pooling with $2 \times 2$ filters and stride 2, and batch normalization, respectively. All performance numbers were measured on a Nvidia GeForce GTX 1080 Ti GPUs and Intel Core i7-6850K CPUs.

A.1 MUTUAL INFORMATION TEST

Below we will introduce more details about tests in Sec. 4.For the numerical task (Shwartz-Ziv & Tishby, 2017), input variable $X$ are 12 binary digits that represent 12 uniformly distributed points on a 2D sphere, and this task is about binary decision rules which are invariant under $O(3)$ rotations of the sphere. $X$ has 4096 different patterns, and they are divided into 64 disjoint orbits of the rotation group, which form a minimal sufficient partition/statistics for spherically symmetric rules (Kazhdan et al., 2003). To generate input-output distribution $P(X, Y)$, Shwartz-Ziv & Tishby (2017) applied a stochastic rule $p(y = 1|x) = \Psi(f(x) - \theta), (x \in X, y \in Y)$, where $\Psi$ is a standard sigmoidal function$\Psi(u) = 1/(1 + exp(-\gamma u))$. Shwartz-Ziv & Tishby (2017) use a spherically symmetric real valued function of the pattern $f(x)$ (evaluated through its spherical harmonics power spectrum (Kazhdan et al., 2003) and compared it to a threshold $\theta$, which was selected to make $p(y = 1) = \sum_x p(y = 1|x)p(x) \approx 0.5$, with uniform $p(x)$. $\gamma$ is high enough to keep the mutual information $I(X; Y) \approx 0.99$ bits. 80% of the data (3277 data pairs) are used for training and rests (819 data pairs) are used for testing. The forward and reverse pass structures of the fully connected neural networks are in Table 7. Hyper parameters used for training are listed in Table 8.

All layer are used in racecar loss. All layers in the $RR_A^6$ and $Std_A$ are reused for training $RR_{AA/AB}^6$ and $Std_{AA/AB}$. For $Ort_A$, we used the Spectral Restricted Isometry Property (SRIP) regularization (Bansal et al., 2018),

$$\mathcal{L}_{SRIP} = \beta\sigma(W^T W - I),\qquad(2)$$

where $W$ is the kernel; $I$ denotes an identity matrix; $\beta$ represents the regularization coefficient; $\sigma(W) = sup_{z \in \mathbb{R}^n, z \neq 0} \frac{\|W_z\|}{\|z\|}$ denotes the spectral norm of $W$. Details about all models' accuracy are shown in Table 1 and Table 2.

Mutual information (MI) plane is a powerful tool for neural networks analysis but admittedly not very intuitive. Here we will use MI plane of $Std_A$ in Fig. 2 as an example to illustrate MI plane in details. From Table 7, we can see that neural network in this numerical task has 5 middle layers ($L_{1\sim5}$) and one output layer ($L_6$). The $X$ axis of the MI plane represents the quantity $I(X; L)$, i..e the mutual information between input variable $X$ and output of each layer $L$. The $Y$ axis of the MI plane represents $I(L; Y)$, the mutual information between output of each layer $L$ and output variable $Y$. Besides, every point in the graphs is colored w.r.t.

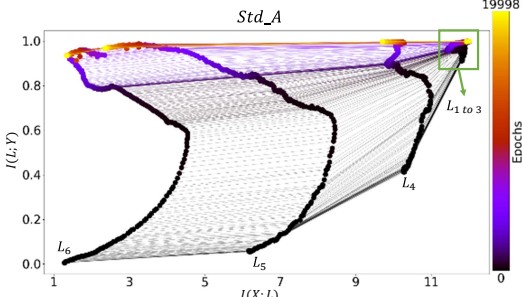

Figure 14: MI plane of $Std_A$ as an example.

the training epochs, i.e., initially black, and yellow once the training is finished. Hence, we can see six lines changing from black to yellow in figure 14 of the updated version. According to the information bottleneck principle [5], the outputs of the early layers contain more information from the input, which means a high value for $I(X; L)$ and $I(L; Y)$. We can see that early layers $L_{1\ 3}$ are located in the top right part of the graph. For later layers, such as $L_6$, parameters of $L_6$ are randomly initialized before training, so there are almost no direct relationships between the output of $L_6$ and $X$ or $Y$, which means low values of $I(X; L)$ and $I(L; Y)$. Once the model is well trained, $L_6$ of $Std_A$ is able to generate data which has the same distribution with $Y$, so $I(L; Y)$ has increased and moved to the top left corner of the graph. The line of $L_6$ starts from the bottom left part of the graph, and moves to the top left part during training. For $RR_A^6$ in figure 2, the L2 constraint is explicitly applied to decrease difference between $L$ and $L'$, and the output is pushed to recover lost information of the input, so $I(X; L)$ of later layers is increased, and $I(X; L)$ of early layers is decreased to make the task easier.

A.2 MNIST CLASSIFICATION

This section gives details for Sec. 5.1. The regular MNIST data set(LeCun et al., 1998) contains $55k$ images for training and $10k$ images for testing. For the n-MNIST motion blur data set(Basu et al., 2017), there are $60k$ images for training and $10k$ images for testing. Image size of them are all $28 \times 28$. Example data from MNIST and n-MNIST with motion blur are shown in Fig. 15. Details

about the forward and reverse pass network structures are shown in Table 9. Hyper parameters are listed in Table 10.

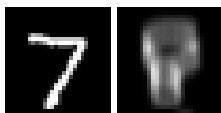

Figure 15: Left: data from MNIST; Right: data from n-MNIST with motion blur.

All 3 convolutional layers are used for racecar loss and all layers in $RR_A^3$, $Std_A$ and $Ort_A$ are reused for training $RR_{AA/AB}^3$, $Std_{AA/AB}$ and $Ort_{AA/AB}$. Example training processes of the MNIST tests are shown in Fig. 16. We can see that racecar loss increases the task difficulty, so $RR_A^3$ yields a lower performance and longer training time than $Std_A$ in the first phase. In the second phase $RR_A^3$ outperforms $Std_A$ and $Ort_A$ in both task $A$ an $B$, which indicates that racecar training is helpful with general feature extraction. For $Ort_A$, we also use equation 2 as orthogonal regularization. All models' accuracy results are listed in Table 3 and Table 4.

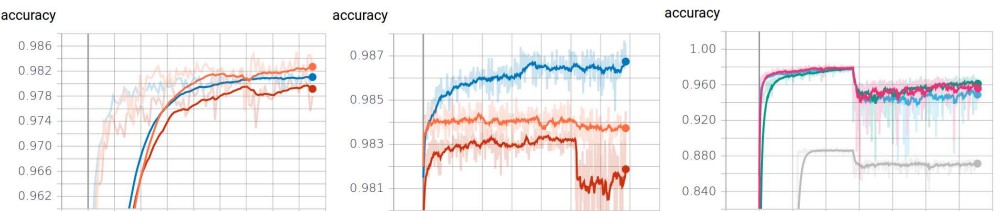

Figure 16: Left: training processes of $RR_A^3$ (blue, accuracy: 0.9810, cost: 5.675 seconds/epoch), $Std_A$ (orange, accuracy: 0.9827, cost: 3.522 seconds/epoch) and $Ort_A$ (red, accuracy: 0.9792, cost: 4.969 seconds/epoch). Middle: training processes of $RR_{AA}^3$ (blue), $Std_{AA}$ (orange) and $Ort_{AA}$ (red). Right: training processes os $RR_{AB}^3$ (green), $Std_{AB}$ (pink), $Ort_{AB}$ (blue), and $Std_B$ (grey).

### A.3 CIFAR TEST

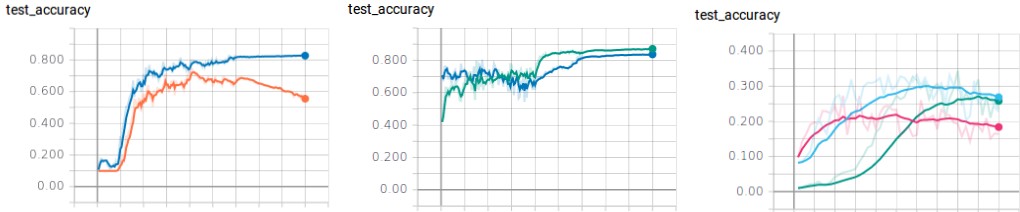

Figure 17: Left: training processes of $RR_A^{13}$ (blue, accuracy: 0.5784, cost: 64 seconds/epoch) and $Std_A$ (orange, accuracy: 0.8272, cost: 63 seconds/epoch). Middle: training processes of $RR_{AA}^{13}$ (green) and $Std_{AA}$ (blue). Right: training processes of $RR_{AB}^{13}$ (blue), $Std_{AB}$ (pink) and $Std_B$ (green).

All 13 convolutional layers are used for racecar loss and all layers except the last fully connected layer (because of different output size) in $RR_A^{13}$ and $Std_A$ are reused for training $RR_{AA/AB}^{13}$ and $Std_{AA/AB}$. Example training processes of the Cifar tests are shown in Fig. 17. In phase II, the model $RR_A^{13}$ outperform $Std_A$ in both task $A$ and $B$. We show details about all models in Table 5.

### A.4 SMOKE DATA SETS TEST

In this section, we will introduce details for Sec. 5.3. The smoke simulation data was generated with a standard fluid solver (Stam, 1999) with MacCormack advection and MiC-preconditioned CG

solver via the $mantaflow$ library. We generated 20 simulations with 120 frames for every simulation. 10% of the data was used for training. Smoke inflow region, inflow velocity and buoyancy force were randomized to produce varied data. The low resolution data was down-sampled from the high-resolution data by a factor of 4. Data augmentation, such as flipping and rotation was used in addition. The smoke capture data set contains 2500 smoke images (Eckert et al., 2018), and we again used 10% as training data set. The forward and reverse pass structures of the network are in Table 13. Hyper parameters are listed in Table 14. Note that inputs of discriminator contain high resolution data $(64, 64, 1)$ and low resolution $(16, 16, 1)$, which is up-sampled to $(64, 64, 1)$ and concatenated with high resolution data.

All generator layers are involved in racecar loss. All 6 layers of $RR_A^6$ and $Std_A$ are reused for training $RR_{AB_1}^6$ and $Std_{AB_1}$, but only 5 layers are reused for training $RR_{AB_2}^6$ and $Std_{AB_2}$ because of different data size. The following results give further details for the auto-encoder transfer learning task $AB_1$ which uses synthetic, i.e., simulated fluid data. Example training processes of $RR_{AB_1}^6$ (orange), $Std_{AB_1}$ (blue) and model trained from scratch $Std_{B_1}$ (red) are shown in Fig. 18. $RR_{AB_1}^6$ achieved the lowest L2 loss after training. Example outputs of $RR_{AB_1}^6$, $Std_{AB_1}$ and $Std_{B_1}$ are shown in Fig. 19. It becomes clear that model $RR_{AB_1}^6$ gives the best performance across these models.

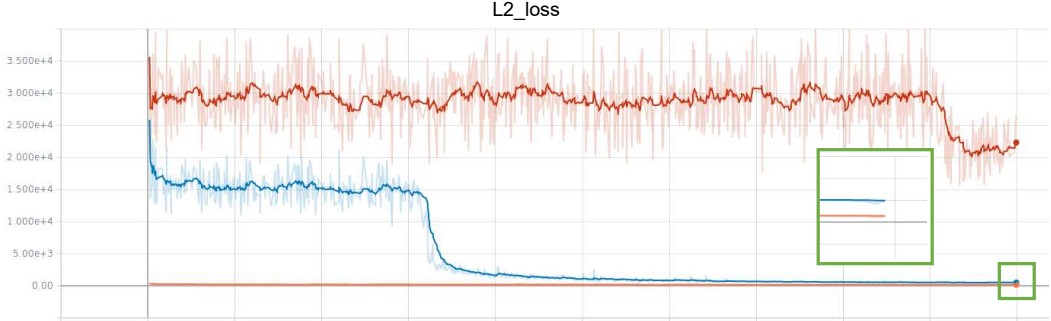

Figure 18: Training processes of $RR_{AB_1}^6$ (orange), $Std_{AB_1}$ (blue) and $Std_{B_1}$ (red). The green inset shows the final loss values.

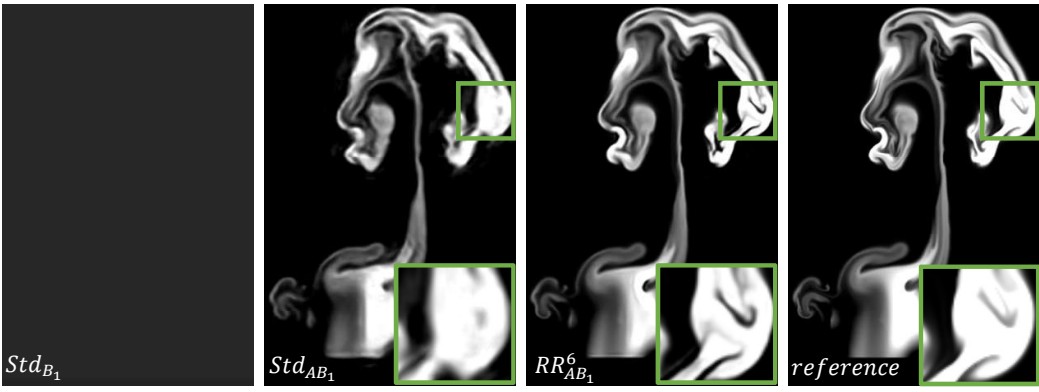

Figure 19: Example outputs comparisons between $RR_{AB_1}^6$, $Std_{AB_1}$, $Std_{B_1}$ and reference. We can see that $RR_{AB_1}^6$ works better than $Std_{AB_1}$, while $Std_{B_1}$ failed for this task producing a mostly black image.

We similarly illustrate the behavior of the transfer learning task $AB_2$ for images of real-world fluids. This example likewise uses an auto-encoder structure. Example training processes of $RR_{AB_2}^6$, $Std_{AB_2}$ and model trained from scratch $Std_{B_2}$ are shown in Fig. 20. $RR_{AB_2}^6$ yields the best performance at the end of training. Example output comparisons of $RR_{AB_2}^6$, $Std_{AB_2}$ and $Std_{B_2}$ are shown in Fig. 21. We can see that error of $RR_{AB_2}^6$ is lower than $Std_{AB_2}$ and $Std_{B_2}$. Detail comparisons between different models are shown in Table 6.

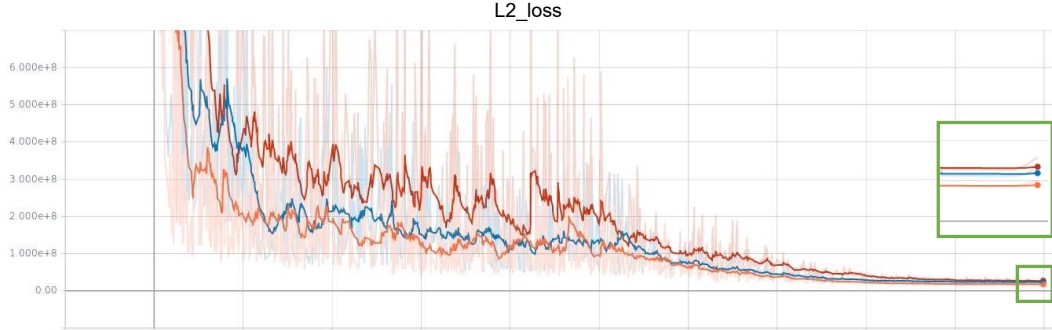

Figure 20: Training processes of $RR_{AB_2}^6$ (orange), $Std_{AB_2}$ (blue) and $Std_{B_2}$ (red). The green inset shows the final loss values.

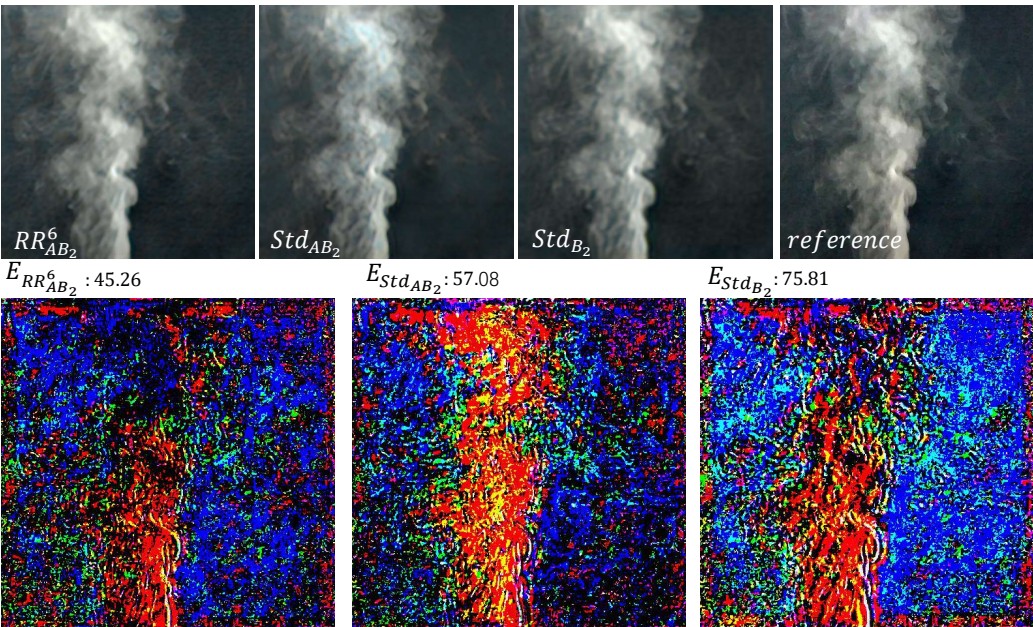

Figure 21: Example outputs and mean absolute error comparisons between $RR_{AB_2}^6$, $Std_{AB_2}$, $Std_{B_2}$ and reference. We can see that error of $RR_{AB_2}^6$ is lower than $Std_{AB_2}$ and $Std_{B_2}$.

### A.5  VGG TEST

We now give details of the tests in Sec. 5.4. For the ImageNet data set (Deng et al., 2009), 1281167 images of 1000 classes are used for training, and 50k images are used for testing. Image size is $224 \times 224$. The forward and reverse pass of VGG19 are in Table 16. Hyper parameters are listed in Table 15.

All 16 convolutional layers are used for racecar loss. To speed up training process of $RR_A^{16}$, we firstly train a model without racecar loss for 6 epochs with batch size 64, as regular training. And then we reuse this model for training $RR_A^{16}$ and $Std_A$ with batch size 24. Example training processes of $RR_A^{16}$ and $Std_A$ are shown in Fig. 22, while those of $RR_{AA}^{16}$ and $Std_{AA}$ are shown in Fig. 23.

Next, we will give additional details for the stylization tests. Gatys et al. (2016) use $L_{total} = \eta L_{content} + \delta L_{style}$ for stylization optimizations, where $\eta$ and $\delta$ are coefficient factors. $L_{content}$ is used to calculate content difference between source image $p$ and generated image $g$, as shown in equation 3.

$$L_{content}(p,g,t) = \tfrac{1}{2} \sum_{m,n} (F_p^{m,n,t} - F_g^{m,n,t})^2, \tag{3}$$

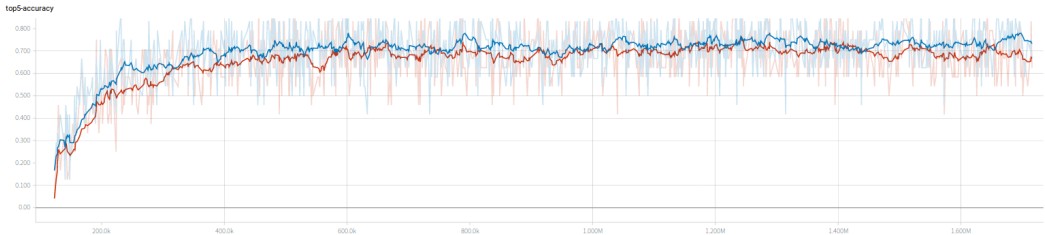

Figure 22: Top5 accuracy of $\text{RR}_A^{16}$ (red, accuracy: 0.6673, cost: 0.636 second/batch) and $\text{Std}_A$ (blue, accuracy: 0.7324, cost: 0.308 second/batch).

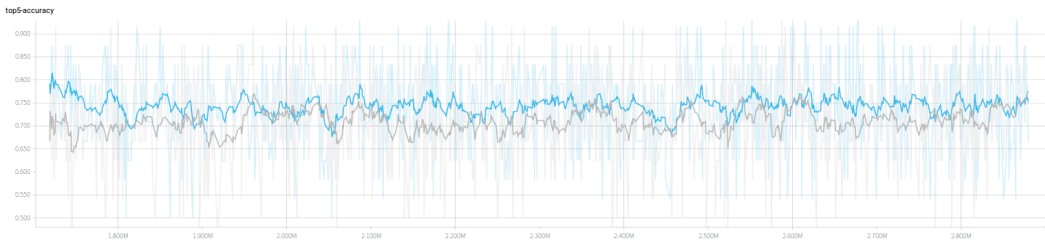

Figure 23: Top5 accuracy of $\text{RR}_{AA}^{16}$ (grey) and $\text{Std}_{AA}$ (blue).

where $F_p^{m,n,t}$ is $p's$ feature representation of the $m^{th}$ filter at position $n$ in layer $t$. $L_{style}$ is used to calculate style difference between style image $a$ and generated image $g$, as shown in equation 4. The style loss can be written as:

$$
\begin{aligned}
G_a^{m,n,t} &= \sum_f F_a^{m,f,t} F_a^{n,f,t}, \\
E_t &= \frac{1}{4N_t^2 M_t^2} \sum_{m,n} (G_g^{m,n,t} - G_a^{m,n,t})^2, \\
L_{style}(a,g) &= \sum_{t=0}^{T} \omega_t E_t,
\end{aligned}
\tag{4}
$$

where $\omega_t$ is are the weighting factors for layer $t$; $G$ is the Gram matrix; $N_t$ denotes filter numbers of layer $t$, and $M_t$ is the dimension of layer $t$'s filter; $T$ denotes the number of layers included in the style loss.

Table 1: Model accuracy of MI base task tests

| training runs | $Ort_A$ | $RR_A^1$ | $RR_A^6$ | $Std_A$ | $Ort_{AA}$ | $RR_{AA}^1$ | $RR_{AA}^6$ | $Std_{AA}$ |
|---|---|---|---|---|---|---|---|---|
| 1 | 0.976 | 0.682 | 0.767 | 0.973 | 0.973 | 0.855 | 0.997 | 0.956 |
| 2 | 0.944 | 0.75 | 0.779 | 0.986 | 0.954 | 0.909 | 0.99 | 0.973 |
| 3 | 0.975 | 0.731 | 0.855 | 0.964 | 0.979 | 0.875 | 0.99 | 0.986 |
| 4 | 0.993 | 0.923 | 0.855 | 0.981 | 0.994 | 0.984 | 0.995 | 0.965 |
| 5 | 0.965 | 0.405 | 0.852 | 0.94 | 0.967 | 0.965 | 0.999 | 0.986 |
| Avg. | 0.971 | 0.707 | 0.822 | 0.967 | 0.973 | 0.938 | **0.994** | 0.973 |
| Std. Dev. | 0.018 | 0.187 | 0.045 | 0.018 | 0.015 | 0.056 | **0.004** | 0.013 |

Table 2: Model accuracy of MI transfer task tests

| training runs | $Ort_{AB}$ | $RR_{AB}^1$ | $RR_{AB}^6$ | $Std_{AB}$ | $Std_B$ |
|---|---|---|---|---|---|
| 1 | 0.978 | 0.914 | 0.996 | 0.136 | 0.966 |
| 2 | 0.948 | 0.956 | 1 | 0.136 | 0.958 |
| 3 | 0.976 | 0.961 | 0.993 | 0.136 | 0.984 |
| 4 | 0.994 | 0.97 | 0.996 | 0.136 | 0.974 |
| 5 | 0.968 | 0.959 | 0.998 | 0.136 | 0.989 |
| Avg. | 0.973 | 0.956 | **0.997** | 0.136 | 0.974 |
| Std. Dev. | 0.017 | 0.022 | 0.003 | **0** | 0.013 |

Table 3: Model accuracy of MNIST base task tests

| training runs | $Ort_A$ | $Std_A$ | $RR_A^3$ | $Ort_{AA}$ | $Std_{AA}$ | $RR_{AA}^3$ |
|---|---|---|---|---|---|---|
| 1 | 0.9792 | 0.9827 | 0.981 | 0.9819 | 0.9854 | 0.9856 |
| 2 | 0.8322 | 0.9815 | 0.9824 | 0.982 | 0.9837 | 0.9851 |
| 3 | 0.9511 | 0.9815 | 0.9828 | 0.9792 | 0.983 | 0.9862 |
| 4 | 0.8841 | 0.9817 | 0.9816 | 0.9814 | 0.9842 | 0.9867 |
| 5 | 0.8266 | 0.9829 | 0.9816 | 0.9816 | 0.9846 | 0.9861 |
| Avg. | 0.895 | 0.982 | 0.982 | 0.981 | 0.984 | **0.986** |
| Std. Dev. | 0.0689 | 0.0007 | 0.0007 | 0.0011 | 0.0009 | **0.0006** |

Table 4: Model accuracy of MNIST transfer task tests

| training times | $Ort_{AB}$ | $Std_{AB}$ | $RR_{AB}^3$ | $Std_B$ |
|---|---|---|---|---|
| 1 | 0.949 | 0.956 | 0.9612 | 0.8714 |
| 2 | 0.8629 | 0.9458 | 0.954 | 0.9657 |
| 3 | 0.9458 | 0.9543 | 0.9648 | 0.9659 |
| 4 | 0.9495 | 0.9559 | 0.9604 | 0.8789 |
| 5 | 0.8564 | 0.9466 | 0.9649 | 0.9638 |
| Avg. | 0.913 | 0.952 | **0.961** | 0.929 |
| Std. Dev. | 0.0485 | 0.0051 | **0.0044** | 0.0494 |

Table 5: Model accuracy of Cifar tests

| training runs | $Std_A$ | $RR_A^{13}$ | $Std_{AA}$ | $RR_{AA}^{13}$ | $Std_{AB}$ | $RR_{AB}^{13}$ | $Std_B$ |
|---|---|---|---|---|---|---|---|
| 1 | 0.8263 | 0.5784 | 0.8351 | 0.869 | 0.2063 | 0.2602 | 0.2581 |
| 2 | 0.7868 | 0.5553 | 0.7936 | 0.8538 | 0.1838 | 0.2687 | 0.2659 |
| 3 | 0.7729 | 0.7686 | 0.7718 | 0.8449 | 0.1795 | 0.2803 | 0.2386 |
| 4 | 0.865 | 0.7382 | 0.8472 | 0.8692 | 0.1738 | 0.2969 | 0.2418 |
| 5 | 0.781 | 0.6775 | 0.7787 | 0.8418 | 0.1704 | 0.2733 | 0.2621 |
| Avg. | 0.806 | 0.664 | 0.805 | **0.856** | 0.183 | **0.276** | 0.253 |
| Std. Dev. | 0.0387 | 0.0945 | 0.0340 | **0.0130** | 0.0141 | 0.0138 | **0.0123** |

Table 6: Model L2 loss of smoke tests. Results for $B_2$: $\times 10^7$.

| training runs | $Std_{AB_1}$ | $RR_{AB_1}^6$ | $Std_{B_1}$ | $Std_{AB_2}$ | $RR_{AB_2}^6$ | $Std_{B_2}$ |
|---|---|---|---|---|---|---|
| 1 | 22883 | 211.9 | 31911 | 213 | 2.95 | 6.13 |
| 2 | 493.3 | 139 | 22291 | 2.37 | 1.79 | 2.66 |
| 3 | 1828 | 210.2 | 31911 | 11.4 | 2.32 | 3.43 |
| Avg. | 8401.43 | **187.03** | 28704.33 | 75.6 | **2.36** | 4.07 |
| Std. Dev. | 12559.15 | **41.61** | 5554.11 | 119 | **0.578** | 1.82 |

Table 7: Forward and reverse pass of the neural network in MI tests

| |
|---|
| Forward pass (294 weights):
$I(12) \rightarrow tanh(FC(10) + b_1) \rightarrow tanh(FC(7) + b_2) \rightarrow tanh(FC(5) + b_3) \rightarrow tanh(FC(4) + b_4) \rightarrow tanh(FC(3) + b_5) \rightarrow tanh(FC(2) + b_6) \rightarrow O(2)$. |
| Reverse pass:
$O(2) - b_6 \rightarrow tanh(FC(3)) - b_5 \rightarrow tanh(FC(4)) - b_4 \rightarrow tanh(FC(5)) - b_3 \rightarrow tanh(FC(7)) - b_2 \rightarrow tanh(FC(10)) - b_1 \rightarrow tanh(FC(12)) \rightarrow I^{'}(12)$. |

Table 8: Hyper parameters of MI tests

| Batch size | 512 | Learning rate | 0.0004 | $\lambda_{1 \sim 6}$ | $1E - 2$ |
|---|---|---|---|---|---|
| Training Epochs | 20000 for $RR^6_{A/AA/AB}$ and $Std_{A/AA/AB}$; 40000 for $Std_B$ | | | | |

Table 9: Forward and reverse pass network of MNIST Classification tests

| |
|---|
| Forward pass (38645 weights):
$I(28, 28, 1) \rightarrow relu(C(3, 64, 1) + b_1) \rightarrow MP \rightarrow relu(C(3, 64, 1) + b_2) \rightarrow MP \rightarrow relu(C(3, 1, 1) + b_3) = I_r \rightarrow FC(10) \rightarrow O(10)$ |
| Reverse pass:
$I_r - b_3 \rightarrow relu(D(3, 64, 1)) \rightarrow UP - b_2 \rightarrow relu(D(3, 64, 1)) \rightarrow UP - b_1 \rightarrow relu(D(3, 1, 1)) \rightarrow I^{'}(28, 28, 1)$ |

Table 10: Hyper parameters of MNIST Classification tests

| Batch size | 64 | $\lambda_{1 \sim 3}$ | $1E - 5$ |
|---|---|---|---|
| Learning rate | 0.001 for $RR^3_A$, $Std_A$ and $Ort_A$;
0.0001 for $RR^3_{AA/AB}$, $Std_{AA/AB/B}$ and $Ort_{AA/AB}$ | | |
| Training Epochs | 100 for $RR^3_A$, $Std_A$ and $Ort_A$;
400 for $RR^3_{AA}$, $Std_{AA}$ and $Ort_{AA}$;
700 for $RR^3_{AB}$, $Std_{AB}$ and $Ort_{AB}$ | | |

Table 11: Forward and reverse pass of the neural network in Cifar tests

| |
|---|
| Forward pass:
$I(32, 32, 3) \rightarrow relu(BN(C(3, 64, 1) + b_1)) \rightarrow relu(BN(C(3, 64, 1) + b_2)) \rightarrow MP$
$\rightarrow relu(BN(C(3, 128, 1) + b_3)) \rightarrow relu(BN(C(3, 128, 1) + b_4)) \rightarrow MP \rightarrow relu(BN(C(3, 256, 1) + b_5))$
$\rightarrow relu(BN(C(3, 256, 1) + b_6)) \rightarrow relu(BN(C(3, 256, 1) + b_7)) \rightarrow MP \rightarrow relu(BN(C(3, 512, 1) + b_8))$
$\rightarrow relu(BN(C(3, 512, 1) + b_9)) \rightarrow relu(BN(C(3, 512, 1) + b_{10})) \rightarrow MP \rightarrow relu(BN(C(3, 512, 1) + b_{11}))$
$\rightarrow relu(BN(C(3, 512, 1) + b_{12})) \rightarrow relu(BN(C(3, 512, 1) + b_{13})) = I_r \rightarrow relu(BN(FC(4096) + b_{14}))$
$\rightarrow relu(BN(FC(4096) + b_{15)}) \rightarrow relu(BN(FC(10) + b_{16)}) \rightarrow O(10)$. |
| Reverse pass:
$I_r - b_{13} \rightarrow relu(BN(D(3, 512, 1))) - b_{12} \rightarrow relu(BN(D(3, 512, 1))) - b_{11} \rightarrow relu(BN(D(3, 512, 1)))$
$\rightarrow UP - b_{10} \rightarrow relu(BN(D(3, 512, 1))) - b_9 \rightarrow relu(BN(D(3, 512, 1))) - b_8 \rightarrow relu(BN(D(3, 256, 1)))$
$\rightarrow UP - b_7 \rightarrow relu(BN(D(3, 256, 1))) - b_6 \rightarrow relu(BN(D(3, 256, 1))) - b_5 \rightarrow relu(BN(D(3, 128, 1)))$
$\rightarrow UP - b_4 \rightarrow relu(BN(D(3, 128, 1))) - b_3 \rightarrow relu(BN(D(3, 64, 1))) \rightarrow UP - b_2$
$\rightarrow relu(BN(D(3, 64, 1))) - b_1 \rightarrow relu(BN(D(3, 3, 1))) \rightarrow I^{'}(32, 32, 3)$. |

Table 12: Hyper parameters of Cifar tests

| Batch size | 200 | $\lambda_{1 \sim 13}$ | $1E - 7$ |
|---|---|---|---|
| Learning rate | 0.1 (0 to 80 epochs);
0.01 (81 to epochs);
0.001 (after 120 epochs) | | |
| Training Epochs | 180 for $RR^{13}_{A/AA}$ and $Std_{A/AA}$;
50 for $RR^{13}_{AB}$ and $Std_{AB/B}$ | | |

Table 13: Forward and reverse pass of network in smoke tests

Generator forward pass:
$I(16, 16, 1) \rightarrow relu(C(5, 64, 1) + b_1) \rightarrow UP \rightarrow relu(C(5, 128, 1) + b_2) \rightarrow UP \rightarrow relu(C(5, 128, 1) + b_3)$
$\rightarrow relu(C(5, 64, 1) + b_4) \rightarrow relu(C(5, 32, 1) + b_5) \rightarrow relu(C(5, 1, 1) + b_6) \rightarrow O(64, 64, 1) = I_r.$

Generator reverse pass:
$I_r - b_6 \rightarrow relu(D(5, 32, 1)) - b_5 \rightarrow relu(D(5, 64, 1)) - b_4 \rightarrow relu(D(5, 128, 1)) - b_3 \rightarrow relu(D(5, 128, 1))$
$\rightarrow MP - b_2 \rightarrow relu(D(5, 64, 1)) \rightarrow MP - b_1 \rightarrow relu(D(5, 1, 1)) \rightarrow I'(16, 16, 1).$

Discriminator:
$I(64, 64, 2) \rightarrow lrelu(BN(C(5, 32, 1) + b_1)) \rightarrow lrelu(BN(C(5, 64, 1) + b_2))$
$\rightarrow lrelu(BN(C(5, 128, 1) + b_3)) \rightarrow lrelu(BN(C(5, 256, 1) + b_4)) \rightarrow FC(1) + b_5 \rightarrow O(1).$

Table 14: Hyper parameters of smoke tests

| Batch size | 64 | Learning rate | 0.0002 | $\lambda_{1\sim6}$ | 0.1 |
|---|---|---|---|---|---|
| Training Epochs | 40000 for $RR_A^6$ and $Std_A$; 1000 for $RR_{AB_1}^6$, $RR_{AB_2}^6$, $Std_{AB_1}$ and $Std_{AB_2}$ | | | | |

Table 15: Hyper parameters of VGG19 training

| $\lambda_{1\sim16}$ | $1E - 10$ |
|---|---|
| Learning rate | 0.01 (0 to 7 epochs); 0.001 (7 to 10 epochs); 0.0001(10 to 15 epochs); 0.00001 (after 15 epochs) |
| Training Epochs | 36 epochs for $RR_A^{16}$ and $Std_A$; 22 epochs for $RR_{AA}^{16}$ and $Std_{AA}$ |

Table 16: Forward and reverse pass of VGG19 network

Forward pass:
$I(224, 224, 3) \rightarrow relu(C(3, 64, 1) + b_1) \rightarrow relu(C(3, 64, 1) + b_2) \rightarrow MP \rightarrow relu(C(3, 128, 1) + b_3)$
$\rightarrow relu(C(3, 128, 1) + b_4) \rightarrow MP \rightarrow relu(C(3, 256, 1) + b_5) \rightarrow relu(C(3, 256, 1) + b_6)$
$\rightarrow relu(C(3, 256, 1) + b_7) \rightarrow relu(C(3, 256, 1) + b_8) \rightarrow MP \rightarrow relu(C(3, 512, 1) + b_9)$
$\rightarrow relu(C(3, 512, 1) + b_{10}) \rightarrow relu(C(3, 512, 1) + b_{11}) \rightarrow relu(C(3, 512, 1) + b_{12}) \rightarrow MP$
$\rightarrow relu(C(3, 512, 1) + b_{13}) \rightarrow relu(C(3, 512, 1) + b_{14}) \rightarrow relu(C(3, 512, 1) + b_{15})$
$\rightarrow relu(C(3, 512, 1) + b_{16}) = I_r \rightarrow MP \rightarrow relu(FC(4096) + b_{17}) \rightarrow relu(FC(4096) + b_{18})$
$\rightarrow relu(FC(1000) + b_{19}) \rightarrow O(1000).$

Reverse pass:
$I_r - b_{16} \rightarrow relu(D(3, 512, 1)) - b_{15} \rightarrow relu(D(3, 512, 1)) - b_{14} \rightarrow relu(D(3, 512, 1)) - b_{13}$
$\rightarrow relu(D(3, 512, 1)) \rightarrow UP - b_{12} \rightarrow relu(D(3, 512, 1)) - b_{11} \rightarrow relu(D(3, 512, 1)) - b_{10}$
$\rightarrow relu(D(3, 512, 1)) - b_9 \rightarrow relu(D(3, 256, 1)) \rightarrow UP - b_8 \rightarrow relu(D(3, 256, 1)) - b_7$
$\rightarrow relu(D(3, 256, 1)) - b_6 \rightarrow relu(D(3, 256, 1)) - b_5 \rightarrow relu(D(3, 128, 1)) \rightarrow UP - b_4$
$\rightarrow relu(D(3, 128, 1)) - b_3 \rightarrow relu(D(3, 64, 1)) \rightarrow UP - b_2 \rightarrow relu(D(3, 64, 1)) - b_1 \rightarrow relu(D(3, 3, 1))$
$\rightarrow I'(224, 224, 3).$

