# OpenReview forum: "Learning General and Reusable Features via Racecar-Training"
_ICLR.cc/2020/Conference — Reject_

### Official Review · AnonReviewer2 · 2019-10-20
**Official Blind Review #2**

**Rating:** 1

**Review:**

This paper attempts to learn general and reusable features for transfer learning tasks. The authors propose a training paradigm called Racecar Training. The core idea of it is to operate a reverse pass for the network. The authors use mutual information to analyze the network for its improved generalizing capabilities. They also conduct experiments on classification, regression and stylization to validate their method’s effectiveness.

Pros:
The reverse pass idea is similar to auto-encoder paradigm that discards redundant information and only save the essential low dimensional one by comparing the original data and the decoding data. The general feature the authors mentioned is like the essential low dimensional information, which is reasonable.

Cons:
1.	I think the main drawbacks of this paper is that the authors make a poor presentation. The authors talk about learning general features with which the model can use on new task in the title, introduction and even the whole paper. However, there are no explicit general features learned during the learning procedure. They only perform the reverse pass when learning in the original task. Even the structure of networks does not change at all. The general and reusable feature is only an explanation of the improved performance. I think the authors should change this explanation to a more convincing one. For example, the network may learn general and reusable weights or parameters since it is the model learned that will be applied to new tasks instead of the features.
2.	The analysis by mutual information makes the paper hard to follow. The figures such as figure 2 are so confusing. There are so many points and lines in each picture. What do they mean? What are the x-axis and y-axis? The authors also do not explain what the meaning of different mutual information are.
3.	The symbols are chaotic. The authors explain “RR^3” means n=1 in equation 1. Then what does n equal to in RR^1? The authors explain “AB” means the model was trained for task A during phase I, and is then trained for task B as transfer in phase II. Then what does AA/AB in “Std_{AA/AB}” mean?
4.	In the last paragraph of page 3, the authors mention orange color. However, there are only green, blue and yellow color in Figure 1.
5.	In experiments, I do not see obvious advantage of the proposed method. For example, in Figure 6, the test accuracy of Std_{AA} is 0.9842 while that of RR^3_{AA} is only 0.9859. In Figure 7, the test accuracy of Std_{AA} is 0.8377 while that of RR^13_{AA} is 0.8711. The transfer tasks (referred as AB) also show minor advantages. With minor advantages but increased requirements for memory and additional computations (e.g., 61.13% slower per epoch for the MNIST tests mentioned by the authors), the proposed method shows very limited values.





**Experience Assessment:**

I have read many papers in this area.

**Review Assessment: Checking Correctness Of Derivations And Theory:**

I carefully checked the derivations and theory.

**Review Assessment: Checking Correctness Of Experiments:**

I carefully checked the experiments.

**Review Assessment: Thoroughness In Paper Reading:**

I read the paper thoroughly.

---

> ### Author Response · Authors · 2019-11-14
> **Reply to reviewer #2**
>
> Dear reviewer, thank you for your review. We are glad you agree that our proposed method can extract general features from the data set. Below we will answer your comments one by one.
>
> --- Regarding presentation: Yes, we agree that what the neural networks learn and what we reused for transfer learning effectively are the parameters and weights of the network. However, the parameters typically represent features in the form of convolutional kernels. For example, when we reuse parameters from a CNN, and this network can extract information from natural image content, such as eyes, then it is common practice to refer to such a feature as an “eye feature”, see, e.g. [1-4].
>
> --- Mutual information (MI) plane visualization: This is a powerful tool for neural networks analysis but admittedly not very intuitive, hence we include more explanations in the corresponding part of the appendix.
> We use the MI plane of Std_{A} in figure 2 as an example to illustrate MI plane in details. The neural network in this numerical task has 5 middle layers ($L_{1\sim5}$) and one output layer ($L_{6}$). The $X$ axis of the MI plane represents the quantity $I(X;L)$, i..e the mutual information between input variable $X$ and output of each layer $L$. The $Y$ axis of the MI plane represents $I(L;Y)$, the mutual information between output of each layer $L$ and output variable $Y$. Besides, every point in the graphs is colored w.r.t. the training epochs, i.e., initially black, and yellow once the training is finished. Hence, we can see six lines changing from black to yellow in figure 14 of the updated version. According to the information bottleneck principle [5], the outputs of the early layers contain more information from the input, which means a high value for  $I(X;L)$ and $I(L;Y)$.
>
> In figure 14 we can see that early layers $L_{1\sim3}$ are located in the top right part of the graph. For later layers, such as $L_{6}$, parameters of $L_{6}$ are randomly initialized before training, so there are almost no direct relationships between the output of $L_{6}$ and $X$ or $Y$, which means low values of $I(X;L)$ and $I(L;Y)$. Once the model is well trained, $L_{6}$ of $Std_A$ is able to generate data which has the same distribution with $Y$, so $I(L;Y)$ has increased and moved to the top left corner of the graph.
>
> In figure 14, we can see that the line of $L_{6}$ starts from the bottom left part of the graph, and moves to the top left part during training. For $RR_{A}^{6}$ in figure 2, the L2 constraint is explicitly applied to decrease difference between $L$ and $L^{'}$, and the output is pushed to recover lost information of the input, so $I(X;L)$ of later layers is  increased, and $I(X;L)$ of early layers is decreased to make the task easier.
>
> --- Typos and symbols, thank you for pointing this out. We corrected the typos and improved the unclear parts.
>
> --- Performance: We repeat tests in figure 3,4,6,7 five times and tests in figure 9 three times. Details about the results are shown in the supplementary materials. The trends and derived conclusions are still consistent with our original submission. We can see that models trained with the racecar loss perform significantly better than standard models.
>
> [1] Heming Liang and Qi Li. Hyperspectral imagery classification using sparse representations of convolutional neural network features. Remote Sensing, 8(2):99, 2016.
> [2] Zetao Chen, Obadiah Lam, Adam Jacobson, and Michael Milford. Convolutional neural network-based place recognition. arXiv preprint arXiv: 1411.1509, 2014.
> [3] Wenyuan Dai, Yuqiang Chen, Gui-Rong Xue, Qiang Yang, and Yong Yu. Translated learning: Transfer learning across different feature spaces. In Advances in neural information processing systems, pp. 353-360, 2019.
> [4] Jason Yosinski, Jeff Clune, Yoshua Bengio, and Hod Lipson. How transferable are features in deep neural networks? In Advances in neural Information processing systems, pp. 3320-3328, 2014.
> [5] Naftali Tishby and Noga Zaslavsky. Deep learning and the information bottleneck principle. In 2015 IEEE Information Theory Workshop (ITW), pp. 1-5. IEEE, 2015.

---

### Official Review · AnonReviewer1 · 2019-11-04
**Official Blind Review #1**

**Rating:** 3

**Review:**

This paper proposes a scheme for training layered feedforward neural networks with backwards accumulation of gradients. In the proposed scheme, the intermediate activations during a forward pass are constrained, using an L2 norm penalty, to be close to the activations of a model that inverts the operations of each layer and transforms the data in reverese order (from output to input). The second, inverse, network shares all parameters with the feedforward network.

The paper hypothesizes that such training, which they call racecar training, results in features that are transferable between tasks; i.e. using racecar training to learn the features, then applying regular training on a novel task. To support this hypothesis, the paper provides an empirical analysis of the mutual information between inputs and intermediate features, and between intermediate features and outputs (as proposed in the information bottleneck literature). Under this analysis, the paper shows that using racecar training the intermediate layers contain less information about the outputs than standard feedforward training. The paper states that this makes the features learned by racecar training better for transfer: features that enable the network to achieve high predictive accuracy on a particular tasks but that carry little information about the output distribution, thus less specialized. The paper continues this analysis to the transfer setting, first to the same initial task, and then the a new task. In both cases one of the variants of the proposed method appears to produce higher accuracy than standard pre-training.

I'm inclining to reject this paper given that the results on the main hypothesis (i.e. transferability of features) seem to provide only marginal improvement, and we have no idea about the repeatability of the results ( how many times did the authors run the experiments for figures 3,4,6,7,9? What's the spread of the results? Are these results significant?). The results on style transfer and super resolution are promising, but these are only illustrative examples: these results do not provide insights on how well the method works in general, or when does it fail.





**Experience Assessment:**

I have read many papers in this area.

**Review Assessment: Checking Correctness Of Derivations And Theory:**

N/A

**Review Assessment: Checking Correctness Of Experiments:**

I assessed the sensibility of the experiments.

**Review Assessment: Thoroughness In Paper Reading:**

I read the paper at least twice and used my best judgement in assessing the paper.

---

> ### Author Response · Authors · 2019-11-14
> **Reply to reviewer #1**
>
> Dear reviewer, we would like to thank you for your comments. We are glad you like our stylization and super resolution results. Below we will answer your comments one by one.
>
> ---- Regarding the repeatability: We repeat tests in figure 3,4,6,7 five times and tests in figure 9 three times. Details about the results are shown in the supplementary materials. The trends and derived conclusions are still consistent with our original submission. We can see that models trained with the racecar loss perform significantly better than standard models.
>
> ---- Regarding the illustrative examples: Since the  mutual information (MI) tests used the MI plane to highlight the properties of our method, we used more illustrative examples in later sections to visually show the impact of our method.

---

### Author Response · Authors · 2019-11-14
**General response to Reviewers**

We would like to thank all reviewers for their comments - they are very important for us in order to improve our paper. We are glad you agree that parts of our results are promising and our proposed method can extract more general features from the data set. To support our conclusion, we have repeated all tests of our original paper, and the summarized performance results will be included in the supplementary materials. Below, we list some of the most important numbers.

To summarize, the key observation is that our approach robustly, over many repeated runs, yields improvements in terms of performance for the original task, and especially for task transfers.
Mutual Information (MI) tests accuracy, 5 runs:
+--------------------Std_{AA}-----RR_{AA}-----Std_{AB}-----RR_{AB}-----+
|       Mean      |      0.973    |      0.994      |    0.136     |       0.997     |
| ___Std. Dev._|____0.013__|____0.004____|__    0.0_  __|__0.003_____|
MNIST tests accuracy , 5 runs:
+--------------------Std_{AA}-----RR_{AA}-----Std_{AB}-----RR_{AB}-----+
|        Mean   |      0.984    |      0.986     |    0.952      |       0.961       |
| __Std. Dev._|___0.0009  |____0.0006__|__0.0051___|____0.0044____|
Cifar tests accuracy, 5 runs:
+--------------------Std_{AA}-----RR_{AA}-----Std_{AB}-----RR_{AB}-----+
|        Mean     |      0.805    |      0.856     |    0.183      |       0.276      |
| __Std. Dev.__|___0.0340__|___0.0130___|__0.0141___|____0.0138__|
Smoke tests L2 loss, 3 runs:
+------------------Std_{AB1}----RR_{AB1}----Std_{AB2}---RR_{AB2}----+
|        Mean     |      8401.4  |      187.0     |    7.56e8      |       2.36e7  |
| __Std. Dev.__|_12559.15__|___41.61 ___|__ 1.19e9     |____5.78e6 _|

We can see that racecar trained models performance significantly better (higher average performance and reduced standard deviation) than models trained with standard procedures. For instance, in MI tests, the accuracy of RR_{AB} is 6.3 times higher than Std_{AB}; for the Cifar tests, the accuracy of RR_{AB} is 50.82% higher than Std_{AB}; for the smoke AutoEncoder tests, the L2 loss of RR_{AB1}  is 300.83 times lower than Std_{AB1}, and the L2 loss of RR_{AB2}  is 31.03 times lower than Std_{AB2}.

---

### Decision · Program_Chairs · 2019-12-19

**Decision:**

Reject

**Comment:**

Both reviewers (we apologize for the lack of a 3rd review) did not feel the paper should be accepted. The rebuttal offered did not change the reviewer scores. So the paper cannot be accepted unfortunately. But the authors should use the feedback to improve their paper and resubmit.